# Peat-Derived ZnCl_2_-Activated Ultramicroporous Carbon Materials for Hydrogen Adsorption

**DOI:** 10.3390/nano13212883

**Published:** 2023-10-31

**Authors:** Egert Möller, Rasmus Palm, Kenneth Tuul, Meelis Härmas, Miriam Koppel, Jaan Aruväli, Marian Külaviir, Enn Lust

**Affiliations:** 1Institute of Chemistry, University of Tartu, Ravila 14a, 50411 Tartu, Estonia; 2Department of Applied Physics, KTH Royal Institute of Technology, SE-106 91 Stockholm, Sweden; 3Institute of Ecology and Earth Sciences, University of Tartu, Ravila 14a, 50411 Tartu, Estonia

**Keywords:** hydrogen storage, microporous carbon materials, zinc chloride activation

## Abstract

Highly microporous adsorbents have been under considerable scrutiny for efficient adsorptive storage of H_2_. Of specific interest are sustainable, chemically activated, microporous carbon adsorbents, especially from renewable and organic precursor materials. In this article, six peat-derived microporous carbon materials were synthesized by chemical activation with ZnCl_2_. N_2_ and CO_2_ gas adsorption data were measured and simultaneously fitted with the 2D-NLDFT-HS model. Thus, based on the obtained results, the use of a low ratio of ZnCl_2_ for chemical activation of peat-derived carbon yields highly ultramicroporous carbons which are able to adsorb up to 83% of the maximal adsorbed amount of adsorbed H_2_ already at 1 bar at 77 K. This is accompanied by the high ratio of micropores, 99%, even at high specific surface area of 1260 m^2^ g^−1^, exhibited by the peat-derived carbon activated at 973 K using a 1:2 ZnCl_2_ to peat mass ratio. These results show the potential of using low concentrations of ZnCl_2_ as an activating agent to synthesize highly ultramicroporous carbon materials with suitable pore characteristics for the efficient low-pressure adsorption of H_2_.

## 1. Introduction

Hydrogen is important for many industrial applications: it is needed as a precursor for synthesis of ammonia and fertilizers, in metal refinement, pharmaceuticals, and, in recent decades, it has been regarded as a possible replacement to traditional fossil fuels as a renewable energy-based energy carrier [1,2,3]. When compared to fossil fuels, it has a higher gravimetric energy density (120 MJ kg^−1^ for H_2_, ~45 MJ kg^−1^ for petrol), and, upon oxidization in a fuel cell, does not generate CO_2_, NO*_x_*, or SO*_x_* compounds into the atmosphere, which are serious greenhouse gases and/or toxic to humans and other living organisms. Owing to these reasons, methods for hydrogen storage are currently under considerable scrutiny [3,4]. While the traditional method of hydrogen storage in pressurized vessels is widely used, it does not come without problems. Hydrogen is not easily compressible, and the high leakage risk that comes with its small molecular size and H_2_ embrittlement of metals under high pressures both cause strict requirements for the storage vessels [4,5]. Even when pressurized to the current industry standard of 700 bar for mobile applications, H_2_ still retains a considerably lower volumetric energy density than gasoline (5.04 MJ dm^−3^ for H_2_ at 273 K vs. 31.68 MJ dm^−3^ for petrol).

For safe and highly efficient H_2_ storage, additional factors to energy density also need to be accounted for. The combination of operational temperature, pressure, and safety concerns due to physicochemical attributes of H_2_ create very specific storage needs for hydrogen. Currently, neither the technology for compressed or cryogenic storage is at a point where either could cost-effectively fill all probable use cases [4,6]. Cryo-adsorption hydrogen storage is a viable solution, utilizing both the benefits of cryogenic temperatures and pressurized storage with a porous adsorbent media for physical adsorption of H_2_ via van der Waals forces. Combined with a low density and high surface area adsorbent, there is potential for high gravimetric density hydrogen storage at relatively low costs. In the spotlight of this research for a viable adsorbent are carbon materials which are a versatile, low-cost candidate with highly tunable pore characteristics [7,8,9,10,11,12,13].

The synthesis of activated carbon materials and their potential applications have been studied extensively in the past by many research groups [9,11,12,13,14]. Activated carbon materials have a high specific surface area (SSA), which is critical for high hydrogen gravimetric densities [8,9,15]. In addition to the SSA, the exact size and geometry of pores play a significant role in the efficient adsorption of hydrogen [8,16,17,18,19,20,21,22]. Ultramicropores, more specifically pores in the width range of 0.6–0.7 nm, offer the highest hydrogen uptake per accessible surface area [8,9,16,19,23,24,25,26]. Pore geometry also plays an important role, as cylindrical and spherical mesopores are shown to help hydrogen confinement [16]. The effect of pore structure-impeding hydrogen diffusion is further advanced by corrugation of the pore walls [17]. Highly porous carbon materials with fine-tuned pore characteristics can be manufactured through several synthesis methods—e.g., by halogenation of carbides for preparation of carbide-derived carbons [8,16,18,27,28], templating inside ordered porous scaffold like a zeolite for zeolite-templated carbons [29,30,31,32], and pyrolysis of organic precursor materials with subsequent chemical and physical activation methods for activated carbons [11,12,13,14,33,34,35,36,37,38,39,40].

While many articles handling pyrolysis of different organic precursors into carbon materials [10,12,34,35,41,42,43,44,45] have been published, peat is a distinctly cheap and easily workable material that has been studied and shows good potential for pyrolysis into carbon materials with different characteristics [14,36,46,47,48,49,50,51]. ZnCl_2_ has been used for chemical activation previously by many groups [10,34,36,39,51,52,53], where both Varila et. al., [36] and Paalo et al. [51] showed that the ZnCl_2_-induced activation of peat precursor is able to yield high SSAs, of over 1200 m^2^ g^−1^, micro and mesoporous carbon materials. Here, we focus on low ZnCl_2_ concentrations to achieve a high ratio of microporosity and minimize the amount of activating compound used.

In this article, peat combined with ZnCl_2_ in six different mass ratios is used to synthesize activated carbon materials. The chemical composition of the resulting carbon materials is determined with X-ray fluorescence (XRF) and energy dispersive X-ray spectrometry (EDS), structure is determined scanning electron microscopy (SEM), crystalline impurity phases with X-ray diffraction (XRD), and porous structure properties are determined with N_2_ and CO_2_ adsorption method. The effectiveness of the used synthesis route for the production of a high SSA microporous carbon material with suitable H_2_ storage properties is investigated and verified by the beforementioned physical characterization methods. Peat is used as a precursor to add value to a relatively cheap and widely available natural resource in Estonia.

## 2. Materials and Methods

### 2.1. Synthesis

The main process steps for the synthesis and characterization of the peat derived activated carbon are shown in Figure 1. Peat from Möllatsi (Tartu County, Estonia) peat deposit and zinc chloride (ZnCl_2_, anhydrous, 99.7% purity, Sigma-Aldrich, Saint-Louis, MO, USA) were mixed in pre-determined mass ratios with ultrapure water (Milli-Q, 18.2 MΩ cm @ 298 K, Merck Millipore, Burlington, MA, USA) as the solvent, where around 7 mL of ultrapure water was added per g of combined peat and ZnCl_2_ mixture. The resulting slurry was stirred until homogenization. Milli-Q water was then evaporated on a hot plate at 360 K and additional drying of the slurry was performed overnight in a Vaciotem TV vacuum oven (J.P Selecta, Barcelona, Spain) at 100 mbar and 360 K. The dried synthesis mixture was transferred in a quartz crucible into a CTF 12/65/550 tube furnace (Carbolite, Derbyshire, UK). Temperature was ramped up 5 K min^−1^ to 973 K and held for 2 h, under a constant Ar gas flow (5.0, Linde, Dublin, Ireland). Synthesized carbon materials were put into a 250 mL flask with dilute 1 M solution of nitric acid (HNO_3_, ≥95%, Honeywell, Charlotte, NC, USA) and stirred on a hotplate for 24 h to remove inorganic impurities. Afterwards, the carbon materials were rinsed with Milli-Q water in the flask and strained through a Büchner funnel equipped with filter paper. The resulting carbon material was dried in a vacuum oven (100 mbar, 360 K) overnight, and weighed.

### 2.2. Physical Characterisation

EVO MA 15 (Zeiss, Oberkochen, Germany) scanning electron microscope (SEM) with 20 kV of accelerating voltage was used to analyze the morphology of synthesized carbon materials. Oxford MAX80 (Oxford Instruments, Abingdon, UK) energy-dispersive X-ray spectrometer integrated into the SEM device was used to determine the presence of inorganic additives. X-ray diffraction (XRD) measurements were performed with a D8 Advance (Bruker, Billerica, MA, USA) X-ray diffractometer with LynxEye detector and with a Ni-filtered Cu Kα radiation source. XRDs of samples were measured in the 2*θ* range 10–90° with a step of 0.013° and a total step time of 172 s on a silicon monocrystal sample holder. For data analysis Diffrac Suite was used with the ICDD PDF4+ 2020 database [54]. Handheld X-ray fluorescence (XRF) spectrometer TRACER 5i (Bruker, Billerica, MA, USA) was used for semi-quantitative determination of select compounds and elements in the carbon material. Primary radiation from a Rh target with a counting time of 180 s was used for the XRF measurements.

### 2.3. Gas Adsorption and Porosity Characterisaion

Gas adsorption measurements with N_2_ (6.0, Linde, Dublin, Ireland) and CO_2_ (5.2, Linde, Dublin, Ireland) were performed with ASAP 2020 and 3Flex (Micromeritics, Norcross, GA, USA) gas sorption analyzers at 77 K and 273 K and up to *p*/*p*_0_ = 0.995 and *p*/*p*_0_ = 0.03, respectively. The samples were degassed before measurements at 573 K and under a vacuum of at least 8 μbar for at least 24 h. Brunauer–Emmett–Teller (BET) theory was used for calculation of SSA [55], and two-dimensional non-local density functional theory for heterogeneous surface (2D-NLDFT-HS) model for carbon materials was used to calculate the pore size distributions (PSD) from simultaneous fitting to N_2_ and CO_2_ isotherm data from which SSA, denoted as *S*_DFT_, and pore volumes, *V*_DFT_, for different pore sizes were obtained [56,57]. SSA of micropores, denoted as *S*_micro_, is calculated as the cumulative surface area up to pore width *w* value of 2 nm. Volume of micropores, *V*_micro_, and volume of ultramicropores, *V*_0.8nm_, are calculated as the cumulative pore volume of pores with widths up to 2 nm and 0.8 nm, respectively.

H_2_ (5.0, Linde, Dublin, Ireland) adsorption measurements of all investigated samples were performed with ASAP 2020 (Micromeritics, Norcross, GA, USA) gas adsorption analyzer at 77 K and up to *p* = 1200 mbar. The samples were degassed before measurements at 573 K and under a vacuum of at least 8 μbar for at least 24 h. The Sips isotherm equation (Equation (1)) was used to fit the H_2_ adsorption data [58].
(1)$$n_{p}=n_{max}\frac{{\left(b\cdot p\right)}^{1/x}}{1+{\left(b\cdot p\right)}^{1/x}}\;$$
where *n*_p_ is the amount of adsorbed gas at pressure *p*, *n*_max_ is the maximum amount of adsorbed gas, *b* is the equilibrium constant, and *x* is a parameter describing the heterogeneity of the adsorbate–adsorbent system, where *x* = 1 characterizes an energetically homogenous adsorption system. All samples were degassed at 573 K and under a vacuum of 16 μbar for at least 12 h prior to gas adsorption measurements.

## 3. Results

A total of six peat-derived activated carbon (PDAC) materials were synthesized with different ZnCl_2_ to peat mass ratios in the initial synthesis mixture (Table 1).

### 3.1. Physical Characterization

Scanning electron microscopy (SEM) revealed that all carbon materials had a heterogenous surface microstructure (Figure A1 in Appendix A). Some structures resembling plant matter were imaged (Figure A1a–d) as the original macroscopic intercellular plant structure is retained and which falls in line with other published SEM images of carbonized plant matter [59,60]. Carbon particles in the size range of 1–15 μm were predominant, with some outlying anomalies of structures sized up to 100 μm. Generally, a spatially uniform distribution of impurities (Figure 2, Figure A2, Figure A3, Figure A4, Figure A5 and Figure A6 in Appendix B) was shown with energy-dispersive X-ray spectroscopy (EDS). The main exception is Si, which is present in all samples as small particles with a width of a few tens of μm. PDAC-2 and PDAC-1 are also outliers, with more concentrated sites of Zn, Al, Si, and S corresponding to small particles (Figure A2 and Figure A3). EDS spectra of all PDACs are brought in Appendix C.

XRF (Figure 3a) and XRD (Figure 3b) were used to further investigate the presence of non-carbon elements and crystalline phases considered as impurities. Several inorganic impurity phases were detected by XRD for all samples (Figure 3b). Diffraction peaks caused by ZnO were the most intense for PDAC with ZnCl_2_ to peat ratio ≤ 0.2. Most intense diffraction peaks corresponding to ZnO were measured in PDAC-0.1, PDAC-0.15, and PDAC-0.2, while other samples exhibited the same diffraction peaks at a much lower intensity. The diffraction peak intensity from ZnS was almost equivalent in all PDAC samples. Diffraction peaks characteristic to SiO_2_, Fe_3_O_4_, and MgO crystalline phases were determined in all samples (Figure 3b). No clear dependency between non-ZnO crystalline phases and the synthesis conditions was determined. PDAC-2 presented itself as an outlier in the series yielding more intense diffraction peaks than PDAC-1 and PDAC-0.5 for SiO_2_, Fe_3_O_4_, and MgO. XRF quantitative results show that PDAC-0.1, PDAC-0.15, and PDAC-0.2 contain relatively high amounts of impurities (Figure 3a, Table 1). PDAC-0.1 exhibits Zn, Fe, Cl, and Ca in amounts over 1 mass%. PDAC-0.15 and PDAC-0.2 yield similar results from XRF, with non-carbon elements making up around or in excess of 1 mass%. PDAC-0.5, PDAC-1, and PDAC-2 present a noticeably lower mass of different impurities, with only Fe content being around 1 mass%. Fe, Al, Si, and S compounds have been shown to be naturally occurring in peat soils [61], which correlates to the results derived from physical characterization of PDAC samples.

### 3.2. Specific Surface Area and Pore Size Distributions

The highest 2D-NLDFT-HS SSA, *S*_DFT_, of 1280 m^2^ g^−1^ was exhibited by PDAC-1 (Table 2). With a higher concentration of ZnCl_2_ in the synthesis mixture, in case of PDAC-2, both the SSA and microporosity decreased—PDAC-2 exhibits a lower *S*_DFT_ of 1020 m^2^ g^−1^ of which only 70% is made up of micropores. In case of a <0.5 ZnCl_2_ to peat ratio, the SSA decreases but the percentage of micropores of the total SSA increases (Table 2). PDAC-0.5 exhibited the largest micropore area, *S*_micro_ = 1250 m^2^ g^−1^, with almost all porosity made up of micropores, *S*_micro_/*S*_DFT_ = 0.99 (Table 2). A more extreme case was present with PDACs prepared at even lower ZnCl_2_ concentrations, where in the case of ZnCl_2_ to peat ratio of ≤0.2, almost all of the porosity is made up of ultramicropores, *V*_0.8nm_/*V*_DFT_ ≥ 0.91 (Table 2). The high degree of ultramicroporosity in PDACs with ZnCl_2_ to peat ratios ≤ 0.2 is showcased in the pore size distributions (PSDs) obtained by fitting the N_2_ and CO_2_ adsorption data simultaneously with the 2D-NLDFT-HS model (Figure 4c). PDAC-2 is the only sample which shows marginal porosity for pores with *w* > 3 nm and the PSD peak describing microporosity shifts towards larger pores with the increase in ratio of ZnCl_2_ to peat used for activation (Figure 4c). Thus, lower quantities of ZnCl_2_ for activation cause the formation of smaller pores and possibly limit the formation of larger pores, giving way to ultramicroporosity.

### 3.3. Hydrogen Adsorption

All PDACs exhibited reversible H_2_ adsorption isotherms without any hysteresis (Figure A13 in Appendix D) at 77 K and, when measured up to 1 bar, where the adsorption and desorption branches of the sorption isotherms match almost perfectly. PDAC-0.5 adsorbed the most H_2_ at 77 K and 1 bar, *n*_H2,1bar_ = 7.54 mmol g^−1^ (Table 3, Figure 4d), while the theoretical maximum according to the Sips isotherm equation (Equation (1)) is achieved by PDAC-1, of *n*_H2,max_ = 12.63 mmol g^−1^. For further analysis, the widely accepted Chahine’s rule can be used as a benchmark, where Chahine’s rule predicts the maximum adsorption excess from the BET SSA values of a material [62]. According to Chahine’s rule a maximum of 1 mass% of H_2_ is adsorbed at 77 K per 500 m^2^ g^−1^ of BET SSA. In case of PDAC-0.1, 1.68 mass% of H_2_ uptake is achieved per *S*_BET_ = 500 m^2^ g^−1^ while a much lower 0.94 mass% is adsorbed for PDAC-2 at the opposite, high ZnCl_2_ concentration end of the series (Table 3). This means PDAC-2 roughly falls into Chahine’s rule while PDAC-0.1 exceeds it by a large margin. Doing the same comparison using *S*_DFT_ instead of *S*_BET_, where the *S*_DFT_ values is considered to be more reliable in case of highly microporous and ultramicroporous carbons [63], the situation is reversed. PDAC-0.1 falls nearly perfectly into Chahine’s rule at 0.99 mass% per 500 m^2^ g^−1^ while PDAC-2 exceeds it somewhat with a value of 1.15 mass% per 500 m^2^ g^−1^ (Table 3). This is due to fact that BET theory underestimates the presence of (ultra)micropores and 2D-NLDFT-HS is a more accurate model for gauging the SSA of (ultra)microporous materials [63]. The adsorption equilibrium constant, *b*, value increases from *b* = 9.38 10^−4^ for PDAC-2 to *b* = 1.83 10^−2^ for PDAC-0.15 (Table 3). The higher *b* value means that a majority of pores are filled with adsorbed H_2_ already at low pressures, i.e., the rate of adsorption is comparatively quick in comparison to that of desorption at 77 K. Whereas the lower *b* value for PDACs with ZnCl_2_ to peat ratio > 0.5 means that desorption rate is higher and higher pressures are required for full surface coverage and pore filling with adsorbed H_2_. This is exemplified by the ratios of *n*_H2,1bar_ to *n*_H2,max_—the ratio of amount of H_2_ adsorbed at 1 bar to the theoretical maximum amount of adsorbed H_2_. This *n*_H2,1bar_/*n*_H2,max_ ratio is highest for PDAC-0.15, at *n*_H2,1bar_/*n*_H2,max_ = 0.833, whereas for PDAC-2 *n*_H2,1bar_/*n*_H2,max_ is only 0.491 (Table 3). This strong H_2_ adsorption and confinement effect is most likely brought forth by the favorable ultramicroporosity present in peat-derived carbon activated with low ratios of ZnCl_2_.

## 4. Discussion

The successful synthesis of microporous carbon materials by pyrolysis from peat and activation with ZnCl_2_ was confirmed based on XRD, SEM, and gas adsorption methods. The obtained carbon materials had similar microstructures and identical impurity compositions of non-carbon elements. The quantity of impurity elements, however, partially depended on the amount of ZnCl_2_ used in the precursor mixture. The compositional analysis results are corroborated by the gas adsorption data, based on which a lower ratio of ZnCl_2_ resulted in a smaller chemical activation effect but a larger ratio of micropores.

Based on the results, ZnCl_2_ as an activating compound initially forms many small (*w* < 1 nm) pores inside the organic precursor matrix and, with an increased ratio of ZnCl_2_ used, with respect to the organic precursor, the formation of wider pores is brought forth. In earlier research conducted by other groups micro-mesoporous activated carbon materials have been successfully synthesized using similar synthesis routes [11,36,51] and ZnCl_2_-to-peat ratios between 2 and 0.5, but the use of ZnCl_2_-to-peat ratios of ≤0.2 have not been published before to the knowledge of the authors. Based on our results and the mesoporous nature of carbon materials synthesized by other groups, it can be theorized that two main mechanisms are at work during ZnCl_2_ activation. Firstly, a higher ZnCl_2_ content acts as a pore former through templating and sterically filling pore-forming spaces and, thus, the initial pores are larger for materials with a higher ZnCl_2_ ratio. Secondly, when larger amounts of ZnCl_2_ are added into the synthesis mixture, the initial smaller pores are fused together during the activation process during which the thermal degradation of ZnCl_2_ removes parts of the underlying carbon microstructure, widening the smaller end of micropores initially present in the porous carbon structure. These two mechanisms help to explain the limited formation of porous structures, where porosity is almost completely made up of micropores (with substantial ultramicroporosity) in case of lower ZnCl_2_ to peat ratios for activation. In addition, they help to explain the formation of an expected high SSA and decrease in microporosity when the ZnCl_2_ to organic material ratio is increased.

The differences in the non-carbon impurity compounds and elements, present after the acid washing process, are also associated with the structural changes sustained in the pyrolysis and chemical activation process. With low ZnCl_2_ ratio, many impurities are encapsulated inside the carbon structure and narrow-necked ultramicropores where they are rendered inaccessible to the acid washing. This applies mainly to Zn-based compounds which are introduced to the forming carbon structure during the chemical activation process—as can be seen from the drastic difference of ZnO present in PDACs with lowest and highest used ZnCl_2_ to peat ratios. With a higher ratio of ZnCl_2_, a much more porous and accessible carbon structure is obtained, thus, leading to a larger accessible surface area, lower microporosity, and lower amounts of non-carbon elements in the chemical composition.

With the increased ratio of ultramicropores to total porosity at low ZnCl_2_ ratios, the H_2_ adsorption equilibrium constant *b* increases rapidly as the additional ultramicropores are able to confine H_2_ very well, especially at low pressures. This is showcased by the high *n*_H2,1bar/_*n*_H2,max_ ratio of 0.833 for PDAC-0.15 (Table 3). At the same time, PDAC-1 which has the highest *n*_H2,max_ = 12.63 mmol g^−1^, has a *n*_H2,1bar/_*n*_H2,max_ ratio of only 0.589, meaning that higher pressures are required for the utilization of the higher H_2_ uptake potential. In addition, the high *n*_H2,1bar/_*n*_H2,max_ ratio of PDACs obtained with low ZnCl_2_ ratios show potential for H_2_ storage at higher temperatures as the high *b* value should be able to support the confinement of H_2_ also in the presence of additional thermal energy. The strong confinement of H_2_ in ultramicropores has been modelled [22,64,65,66] and shown before by different methods [16,25,67,68,69]. The strong confinement of H_2_ in ultramicropores and hydrogen uptakes of up to 10.7 mmol g^−1^ [22] and up to 18 mmol g^−1^ [26] at 77 K and at 1 bar of H_2_ pressure have been shown for activated carbon fiber ACF-15 and for carbons derived from semi-cycloaliphatic polyimide, respectively. Even as the gravimetric amount of adsorbed H_2_ at 1 bar and at 77 K is considerably higher than that exhibited by PDACs, 7.54 mmol g^−1^ for PDAC-0.5, the beforementioned ultramicroporous adsorbents [22,26] do not present a H_2_ isotherm reaching a plateau near 1 bar of pressure and, thus, they do not exhibit as high *n*_H2,1bar_/*n*_H2,max_ ratio as PDAC-s with low ZnCl_2_ ratios. In addition, the gravimetric amount of adsorbed H_2_ is decreased in the case of PDACs by the addition of dead mass from the impurities. Thus, the synthesis of highly ultramicroporous biomass-derived carbons is presented for the strong confinement of H_2_ at low H_2_ pressures. This is enhanced by the reversible H2 uptake and desorption, where no hysteresis was determined from the H2 adsorption isotherms, meaning that this strong confinement of H_2_ a physical adsorption process. The possibility to adsorb H_2_ at low pressures is of specific interest for the development of cryo-adsorptive storage systems able to operate at increased temperatures. Thus, in future, the low-pressure adsorption of H_2_ at elevated *T*s will be investigated to determine the adsorption enthalpy of H_2_ in investigated ultramicropores and determine the viability of such systems for technical H_2_ storage solutions.

In this work we showcase the use of low ratios of ZnCl_2_ as an activating agent for the synthesis of highly ultramicroporous peat-derived carbon materials and the strong H_2_ confining capability of these ultramicroporous adsorbents, especially at lower pressures. The overall lower cost and subsequent environmental impact of using lower amounts of activating agents, e.g., ZnCl_2_, combined with high ultramicroporosity for H_2_ adsorption is a fruitful combination for further research into H_2_ storage materials and systems.

## Figures and Tables

**Figure 1 nanomaterials-13-02883-f001:**
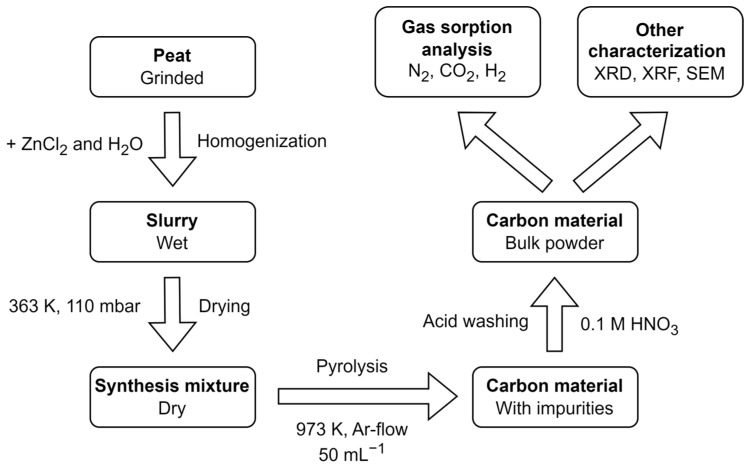
Block scheme of the synthesis route.

**Figure 2 nanomaterials-13-02883-f002:**
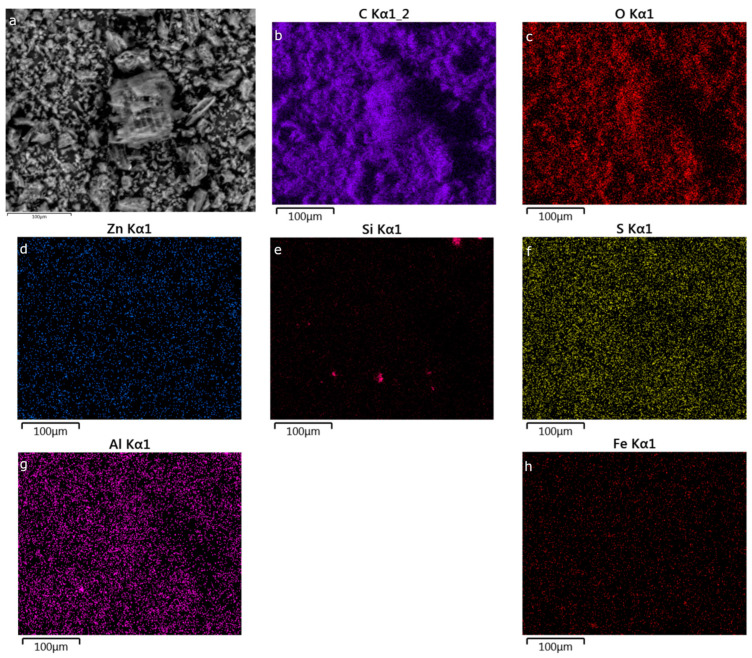
SEM-EDS results of PDAC-0.1 depicting the original SEM image (**a**) and the elemental mapping results of the same location for C, O, Zn, Si, S, Al, and Fe (images (**b**–**h**), respectively).

**Figure 3 nanomaterials-13-02883-f003:**
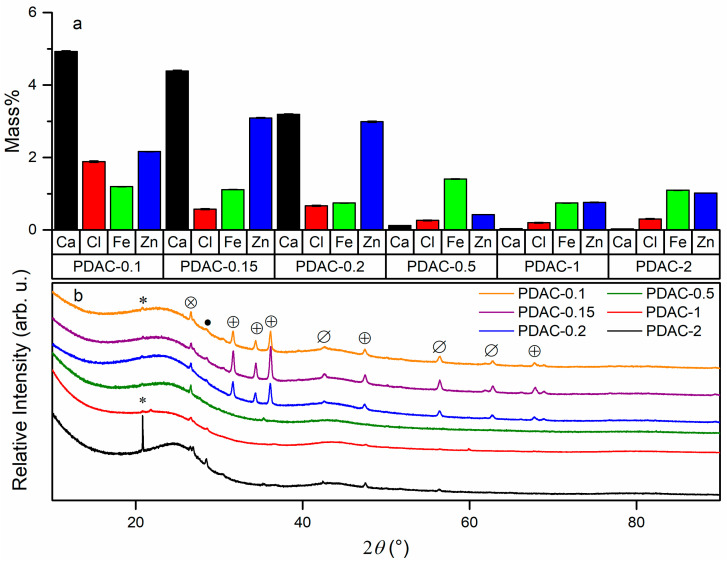
X-ray fluorescence results for select elements (**a**) and diffractograms of all carbon samples (**b**). *, ● and Ø mark diffractogram peaks for SiO_2_, Fe_3_O_4_, and MgO, respectively. ⊗ and ⊕ represent peaks for ZnS and ZnO, respectively. The XRD patterns have been shifted up for better separation and visualization.

**Figure 4 nanomaterials-13-02883-f004:**
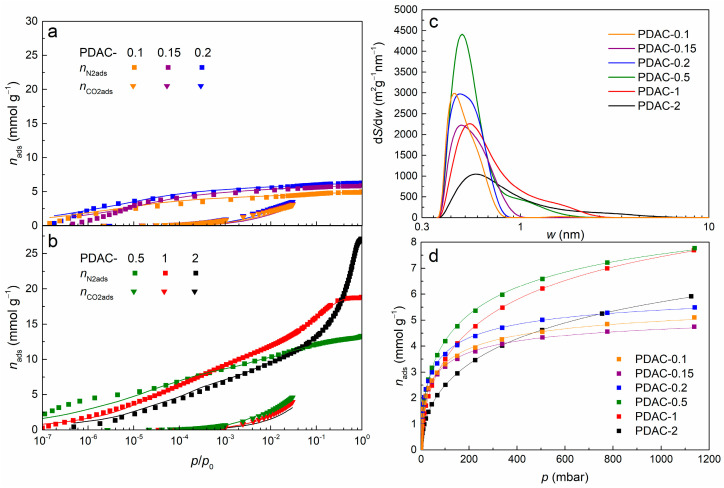
N_2_ and CO_2_ adsorption isotherms (symbols) with accompanying fits (lines) to the 2D-NLDFT-HS model [56,57] for carbon materials simultaneously applied to both N_2_ and CO_2_ adsorption data for PDAC-0.1, PDAC-0.15, PDAC-0.2 (**a**) and PDAC-0.5, PDAC-1, PDAC-2 (**b**). Pore size distributions by differential specific surface area of all carbon materials calculated with SAIEUS software by applying the 2D-NLDFT-HS model [56,57] simultaneously to the N_2_ and CO_2_ isotherm data (**c**). Hydrogen adsorption isotherms (symbols) of all PDAC samples and fits (lines) to the Sips isotherm [58] equation (Equation (1)) (**d**).

**Table 1 nanomaterials-13-02883-t001:** Synthesized peat-derived carbons (PDAC) main primary characteristics for quick guidance. The *x* in PDAC-*x* is the ZnCl_2_ to peat mass ratio.

Carbon	ZnCl_2_:Peat Mass Ratio ^a^	Zn Mass% ^b^	*S*_BET_ (m^2^ g^−1^) ^c^
PDAC-0.1	1:10	2.16 (1)	370
PDAC-0.15	1:7	3.09 (2)	380
PDAC-0.2	1:5	2.99 (3)	470
PDAC-0.5	1:2	0.43 (1)	1030
PDAC-1	1:1	0.76 (2)	1400
PDAC-2	1:0.5	1.02 (1)	1240

^a^ precursor mixture mass ratios, ^b^ the mass% of Zn in the synthesized PDACs obtained from XRF and the standard deviations of determined Zn mass% brought in the brackets corresponding to the last shown decimal, ^c^ BET SSAs of synthesized PDACs.

**Table 2 nanomaterials-13-02883-t002:** Porous structure characteristics of synthesized carbon materials from gas adsorption data.

Carbon Sample	*S*_DFT_ (m^2^ g^−1^)	*S*_micro_ (m^2^ g^−1^)	*S*_micro_/*S*_DFT_ (%)	*V*_DFT_ (cm^3^ g^−1^)	*V_micro_* (cm^3^ g^−1^)	*V*_0.8nm_ (cm^3^ g^−1^)
PDAC-0.1	630	630	100	0.177	0.173	0.161
PDAC-0.15	680	680	100	0.201	0.200	0.185
PDAC-0.2	790	790	100	0.221	0.220	0.212
PDAC-0.5	1260	1250	99	0.446	0.428	0.270
PDAC-1	1280	1220	95	0.608	0.539	0.207
PDAC-2	1020	720	71	0.879	0.340	0.109

*S*_DFT_—Total SSA calculated with 2D-NLDFT-HS model; *S*_micro_—Micropore SSA of pores with *w* < 2 nm, calculated with 2D-NLDFT-HS model; *V*_DFT_, *V_micro_*, *V*_0.8nm_—Total, micropore and ultramicropore volume calculated with 2D-NLDFT-HS model, where 2D-NLDFT-HS model was applied simultaneously to N_2_ and CO_2_ adsorption data in case of all presented parameters [56,57].

**Table 3 nanomaterials-13-02883-t003:** Hydrogen adsorption characteristics of synthesized carbon materials along with Sips isotherm [58] equation (Equation (1)) parameters.

Carbon Sample	*b* (mbar^−1^)	*x*	*n*_H2,max_ (mmol g^−1^)	*n*_H2,max_/*S*_BET_ (Mass%/500 m^2^)	*n*_H2,max_/*S*_DFT_ (Mass%/500 m^2^)	*n*_H2,1bar_ (mmol g^−1^)	*n*_H2,1bar_/*n*_H2,max_
PDAC-0.1	1.35 × 10^−2^	1.86	6.18	1.68	0.99	4.96	0.803
PDAC-0.15	1.83 × 10^−2^	1.81	5.58	1.48	0.83	4.65	0.833
PDAC-0.2	1.72 × 10^−2^	1.79	6.49	1.39	0.83	5.39	0.831
PDAC-0.5	4.12 × 10^−3^	1.82	10.99	1.08	0.88	7.54	0.686
PDAC-1	1.87 × 10^−3^	1.74	12.63	0.91	0.99	7.44	0.589
PDAC-2	9.38 × 10^−4^	1.82	11.61	0.94	1.15	5.70	0.491

*b*—equilibrium constant from the Sips equation; *x*—heterogeneity parameter from the Sips equation; *n*_H2,max_—maximum theoretical H_2_ adsorption according to the Sips equation; *n*_H2,max_/*S*_BET_ and *n*_H2,max_/*S*_DFT_—ratio between maximum mass% of stored H_2_ in carbon material vs. 500 m^2^ of *S*_BET_ or *S*_DFT,_ respectively, for comparison with Chahine’s rule; *n*_H2,1bar_—H_2_ adsorption at 77 K and 1 bar; *n*_H2,1bar_/*n*_H2,max_—ratio of H_2_ adsorption at 77 K 1 bar and the theoretical maximum from Sips equation.

## Data Availability

The data presented in this study are available on request from the corresponding author.

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
