# Peer review of "Peat-Derived ZnCl2-Activated Ultramicroporous Carbon Materials for Hydrogen Adsorption"

_nanomaterials, 2023, doi:10.3390/nano13212883_

Round 1

Reviewer 1 Report

Comments and Suggestions for Authors

The manuscript titled “Peat-derived ZnCl2 Activated Ultramicroporous Carbon Materials for Hydrogen Adsorption” studied the synthesis, activation and characterization of microporous carbon materials for gas adsorption like H2. This work is not well presented and the originality is not clear.

However, major mandatory revision is requested for possible reconsideration in the journal.

1)      The novelty of this work should be cited clearly, in the abstract. Only the most interesting results could be mentioned in the abstract.

2)      The section “Methods and materials should be improved by separate the methods to the materials.

3)      Figure 1: The block scheme of the synthesis route should separate as a scheme on the materials methods. While, the SEM image could be moved on the section “Results and discussion”

4)      Details (instruments, samples, saturation etc…) on the H2 adsorption could be added in the corresponding section.

5)      Results & Discussions section should be improved by interpreting the chemistry involved for adsorption of H2. Comparison on the obtained results with other published work should be added.  Mechanism should be illustrated for more details.

Reviewer 2 Report

Comments and Suggestions for Authors

Dear authors,

please complete the study with results concerning the the number of cycling of gas adsorption and desorption and the percent of adsorption capacity retained by your material after a certain number of cycles. Another important aspect will be to fully discuss the gas release procedure as well the potential usability of these type of materials (like amount of gas adsorbed per gram of material).

Reviewer 3 Report

Comments and Suggestions for Authors

Manuscript ID: nanomaterials-2666290

Title: Peat-derived ZnCl2 Activated Ultramicroporous Carbon Materials for Hydrogen Adsorption

The work reports the synthesis of microporous carbon by chemical activation by ZnCl2. The research demonstrated that using a specific ZnCl2:peat ratio (1:2) it is possible to obtain a high surface area carbon material consisting mainly of micropores. These pore characteristics are suitable for low-pressure adsorption of H2.

The manuscript is well written and the information are presented clearly. Even if I’m not very involved in this field, I found the manuscript clear with well described data.

The characterization analyses are complete and described in deep, and also they give a right support of the conclusion. For all these reasons I guess that this paper is suitable for publication, but I just report below a couple of minor issues.

1) in the Materials and methods:

When you explain the synthesis you report “deposit and zinc chloride (ZnCl2, anhydrous, 99.7% purity, Sigma-Aldrich) were mixed in pre-determined mass ratios with ultrapure water”, a part from the peat:ZnCl2 that is described,  which is the amount of water usually used?

2) in the table 1 of the Results: not very clear what the numbers (1)(2) in brackets in the column of Zn mass % refer to.

Best regards

Reviewer 4 Report

Comments and Suggestions for Authors

The work by Möller et al presents the synthesis and characterization of a potential material for hydrogen storage. Hydrogen storage is a relevant topic. The use of peat as a cheap and sustainable material is appealing. The motivation of chemical activation by the use of ZnCl2 to achieve an optimal balance between specific surface area and very small pores is clearly stated. The experimental results are analyzed in detail in the discussion section. For these reasons, the article is suitable to be published in the journal Nanomaterials. However, some aspects should be amended before being accepted.

-In figure 1(a) and the graphs from figure 4, the letters and numbers are too small and it is difficult to read them. Please, increase the font size.

-The EDS specter from figure 2 and the EDS spectra in the Appendix A are too small and what is written is illegible. Please, increase the font size.

-I kindly ask the authors to include a Conclusion section, where they summarize the most important results and points from the present work. Also, where they explain future ideas about how to continue the work to improve further the performance of the material, and if they plan to use it in a specific device, setup or application. What would be the potential and impact of these results for other researchers and applications in the field?

Comments on the Quality of English Language

The work is well written and the discussion and explanations can be well followed.

Round 2

Reviewer 1 Report

Comments and Suggestions for Authors

As the principal subject of this work is the utilization of Activated Ultramicroporous Carbon for Hydrogen Adsorption;

1)      more discussion on the adsorption behaviors (the mechanism) should be discussed in details. If the mechanism was already discussed, the authors could explain the novelty of this work, particularly, the adsorption of H2.

2)       Also, the authors could compare the own results with other published elsewhere to show the originality of this work.

3)      What about the saturation of the materials and its reusability?

4)      If the authors assume that mechanism and discussion on Hydrogen adsorption are not necessary for this work, the title could change the title and focalized on the theoretical study of hydrogen adsorption. Because to refer some mechanism is not recommended and it was easy for the readers of this journal to understand the phenomena without reading other work.

Reviewer 4 Report

Comments and Suggestions for Authors

The authors have addressed my questions and as I can see through the second reading of the whole manuscript, they also have improved it from the other reviewers suggestions. Therefore, it is ready to be accepted. I just would like to say that the figure 4 appears twice, but I guess that the authors will amend this during the proof reading if published.

Comments on the Quality of English Language

It is clearly written.

Round 3

Reviewer 1 Report

Comments and Suggestions for Authors

The revised manuscript can be accepted